# Structure of the NS2B-NS3 protease from Zika virus after self-cleavage

Wint Wint Phoo[1,2,3,*], Yan Li[4,*], Zhenzhen Zhang[1,3], Michelle Yueqi Lee[4], Ying Ru Loh[4], Yaw Bia Tan[1,3], Elizabeth Yihui Ng[4], Julien Lescar[2,3], CongBao Kang[4] & Dahai Luo[1,2,3]

The recent outbreak of Zika virus (ZIKV) infections in the Americas represents a serious threat to the global public health. The viral protease that processes viral polyproteins during infection appears as an attractive drug target. Here we report a crystal structure at 1.84 Å resolution of ZIKV non-structural protein NS2B-NS3 protease with the last four amino acids of the NS2B cofactor bound at the NS3 active site. This structure represents a post-proteolysis state of the enzyme during viral polyprotein processing and provides insights into peptide substrate recognition by the protease. Nuclear magnetic resonance (NMR) studies and protease activity assays unravel the protein dynamics upon binding the protease inhibitor BPTI in solution and confirm this finding. The structural and functional insights of the ZIKV protease presented here should advance our current understanding of flavivirus replication and accelerate structure-based antiviral drug discovery against ZIKV.

[1] Lee Kong Chian School of Medicine, Nanyang Technological University, EMB 03-07, 59 Nanyang Drive, Singapore 636921, Singapore. [2] School of Biological Sciences, Nanyang Technological University, 60 Nanyang Drive, Singapore 636921, Singapore. [3] NTU Institute of Structural Biology, Nanyang Technological University, EMB 06-01, 59 Nanyang Drive, Singapore 636921, Singapore. [4] Experimental Therapeutics Centre, Agency for Science, Technology and Research (A*STAR), 31 Biopolis way, Nanos, #03-01, Singapore 138669, Singapore. * These authors contributed equally to this work. Correspondence and requests for materials should be addressed to C.K. (email: cbkang@etc.a-star.edu.sg) or to D.L. (email: luodahai@ntu.edu.sg).

Zika virus (ZIKV), a mosquito-borne flavivirus, has triggered a recent global public health crisis leading the World Health Organization to declare ZIKA a global emergency in 2016. Growing evidence links ZIKV infection to fetal microcephaly and neurologic complications in adults such as Guillain-Barré syndrome, acute myelitis and menin-goencephalitis[1–4]. Besides ZIKV, several flaviviruses are important human pathogens including dengue virus (DENV), West Nile virus (WNV), yellow fever virus (YFV), Japanese encephalitis virus (JEV) and tick-borne encephalitis virus (TBEV)[5,6]. There is an unmet need for specific antiviral therapeutics against ZIKV and related pathogenic flaviviruses.

Flaviviruses have a single-stranded positive-sense RNA genome which encodes a polyprotein of about 3,000 amino-acids. During viral replication, this polyprotein is processed into three structural proteins (capsid, membrane and envelope proteins) that are involved in viral particle assembly and seven nonstructural (NS) proteins (NS1, NS2A, NS2B, NS3, NS4A, NS4B and NS5) responsible for viral replication, virion assembly and evasion from the host defence mechanisms[5]. NS2B-NS3 protease is responsible for all cytoplasmic cleavages including at junctions between NS2A/NS2B, NS2B/NS3, NS3/NS4A and NS4B/NS5 proteins and within the capsid, NS2A and NS4A proteins[5,6]. Similar to NS3-NS4A protease from hepatitis C virus, the flavivirus NS2B-NS3 protease is essential for the virus replicative cycle, and thus constitutes an ideal target for antiviral drug development[7]. NS2B-NS3 protease also suppresses the immune response by cleaving stimulator of interferon genes (STING) in DENV (refs 8,9), triggers apoptosis via activating caspases in WNV (ref. 10), and induces neurotropic pathogenesis by inhibiting activator protein 1 (AP-1) in JEV (ref. 11). The NS3 N-terminal domain is a chymotrypsin-like serine protease with an absolutely conserved catalytic triad His51, Asp75 and Ser135, while membrane bound NS2B serves as a cofactor essential for folding and catalysis[12,13]. Crystal structures of several flavivirus proteases were reported in free-enzyme form and in complex with inhibitors, all with a flexible linker covalently tethered between NS2B cofactor peptide (residues 49–97) and NS3 protease domain[7].

Here, we captured the active enzyme in one of its native states by determining a crystal structure of a heterodimeric NS2B-NS3 protease from ZIKV in complex with its own NS2B C-terminal peptide. The structure provides the molecular basis for substrate recognition by the protease. Protein dynamics study using solution NMR confirmed the binding mode of the protease to its peptide substrate and inhibitor (BPTI). We performed functional protease activity assay to demonstrate that the linker between NS2B-NS3 affects the substrate accessibility. Our study provides structural and functional insights into the ZIKV protease and will facilitate antiviral development targeting this enzyme.

## Results

**Structure of ZIKV NS2B-NS3 protease with NS2B C-terminus.** We designed a novel construct of ZIKV NS2B-NS3 protease by removing the hydrophobic regions (residues 1–44 and 97–125) of NS2B from the native NS2B-NS3 protease (containing residues 1–177 of NS3) (this construct is hereafter named 'eZiPro' to highlight the enzymatic cleavage site) (Fig. 1a; Supplementary Fig. 1). We observed that the NS2B-NS3 junction was cleaved during overexpression in *Escherichia coli* and throughout purification steps (Supplementary Fig. 2). We crystallized eZiPro and determined its structure at 1.84 Å resolution (Fig. 1b; Table 1). The eZiPro structure adopts the 'closed conformation' also found in other flaviviral proteases bound to inhibitors, where

the NS2B cofactor encircles the NS3 protease domain (Fig. 1b; Supplementary Fig. 3)[14–18]. We also observed a NS2B-NS3 dimer interface, which is unique to ZIKV and is mediated by a set of polar contacts from both NS2B cofactor and NS3 (Supplementary Fig. 4). The dimerization may be further driven by the membrane association of the polyprotein in the *in vivo* membrane-rich environment where the effective local concentration of the replicative enzymes is very high. Indeed, full length trans-membrane protein NS2B dimerizes in a cell-based assay[19]. N-terminal residues V49-D58 from the NS2B N-terminal region form a β-strand within the N-terminal lobe of the NS3 protease, while C-terminal residues S71-V87 of NS2B fold into a β-hairpin that inserts into the catalytic site of the protease, and participate in binding the substrate. Remarkably, the NS2B C-terminal tetra-peptide T127-G128-K129-R130 makes specific interactions with both the NS3 protease and the remainder of the NS2B cofactor (Fig. 1). Notably, the side chain of R130 extends into the S1 pocket by stacking with Y161 and forming hydrogen bonds with D129 and Y130 from NS3 protease. The carboxylic end of NS2B is stabilized by H51, G133 and S135 from the protease. K129 from NS2B forms hydrogen bonds with S81 and D83 (unique to ZIKV) within the S2 pocket. The S3 pocket is only partially filled due to the short side chain of G128 from NS2B and T127 forms no interaction with the protease but rather projects out of the catalytic pocket. Overall, the L-shaped proteolytic product 'TGKR' peptide is seen to bind to pockets S1, S2 and only partially to S3 of the protease via mostly polar interactions. Interestingly, the presence of a small residue at the P3 position of NS2B from ZIKV (G128), which is unique to ZIKV, leaves the S3 pocket largely unoccupied, offering opportunities for structure-based drug design starting from the natural peptide substrate 'TGKR' (Fig. 1; Supplementary Figs 1 and 5). To confirm that the 'TGKR' peptide remains covalently bound to the rest of NS2B cofactor, we determined the molecular weight of the NS2B and NS3 fragments from the purified eZiPro to be exactly 6.7 kDa and 19.0 kDa, demonstrating that no further degradation of NS2B beyond the NS2B/NS3 junction occurs (Supplementary Fig. 2). No electron density is visible for the first 18 amino acids of NS3 protease, indicating that after cleavage the N-terminal region of NS3 protease falls off from its own S' pocket. Overall, the structure of eZiPro captures the post cleavage state of NS2B-NS3 protease.

**Structural dynamics of ZIKV protease in solution.** To better understand the structural dynamics of eZiPro, we carried out solution NMR studies. We first assigned the $^1$H-$^{15}$N-HSQC spectrum of eZiPro at 37 °C (Supplementary Fig. 6), which indicates that the secondary structures elements of eZiPro in solution are similar to those found in the crystal structure (Supplementary Fig. 7). We then collected the $^{15}$N $T_1$, $T_2$, hetNOE parameters and found that residues including S1-E17 and R170 to E177 of the N- and C-termini of NS3pro are highly dynamic in solution, characterized by low $T_1$ and hetNOE values ($<0.6$) (Fig. 2a). Conversely, residues S48 to D75 of NS2B appear stable in solution, as demonstrated by hetNOE values larger than 0.6 (Fig. 2a). Comparable $T_1$ and $T_2$ values of the NS2B cofactor region to those of NS3pro strongly indicate that NS2B forms a closed form complex with NS3 protease in solution. Residues V76-V87 of NS2B which form a β-hairpin in the crystal structure, are relatively flexible in solution and exhibit relatively broaden peaks, suggesting a dynamic sampling process of the NS2B cofactor binding to NS3 and forming the P2 pocket (Fig. 1b; Supplementary Fig. 6).

**Comparison of different ZIKV protease constructs.** To determine whether the NS2B 'TGKR' peptide occupies the

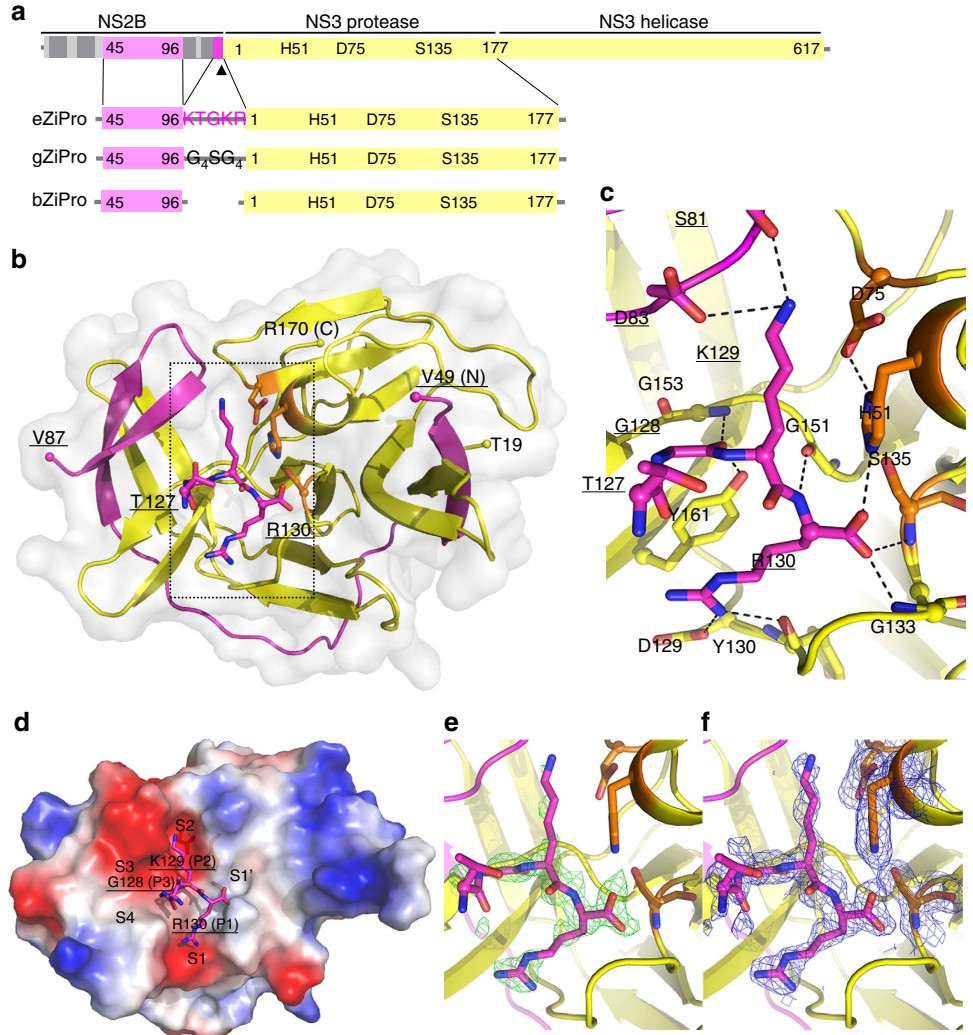

**Figure 1 | Crystal structure of eZiPro in complex with the C terminal TGKR tetra-peptide of NS2B.** (**a**) Full length NS2B and NS3 proteins followed by construct designs for eZiPro, gZiPro, and bZiPro. ZIKV NS3 protease (S1-E177) is covalently linked to NS2B cofactor residues (T45-E96) via K126-R130 of NS2B in eZiPro construct and by $G_4SG_4$ linker in gZiPro construct. pETDUET vector with two promoter sites were used for bZiPro resulting in unlinked bZiPro construct. (**b**) Overall structure of eZiPro showing the TGKR NS2B peptide bound in substrate binding site. NS2B is coloured in magenta and NS3 in yellow. N-terminal residues, C-terminal residues, and residues of the TGKR peptide are shown. (**c**) Close-up views of the interactions between viral peptide and residues from protease. Hydrogen bonds are shown as dashes. (**d**) Surface charge density view of the complex. Substrate binding pockets are labelled. (**e**) A simulated annealing omit map of the TGKR peptide is contoured at $3\sigma$ in green mesh. (**f**) $2mF_o\text{-}DF_c$ electron density map contoured at $1\sigma$ in blue.

protease active pocket in solution and causes steric hindrance to a peptide substrate, as observed in the crystal structure, we designed two more protease constructs: one with a $G_4SG_4$ linker replacing K126-R130 of NS2B which has been commonly used for other flaviviruses (hereafter named gZiPro) leading to a single-chain NS2B-NS3 protease and one bivalent protease consisting of two separate polypeptide fragments 45–96 of NS2B and 1–177 of NS3 (hereafter named bZiPro) (Fig. 1a; Supplementary Fig. 2). We prepared the gZiPro and bZiPro proteins following the same protocol as eZiPro and found that the purified proteins carry correct molecular weight as expected (Supplementary Fig. 2). All three proteases fold as monomeric globular proteins in solution as shown by size exclusion chromatography (Supplementary Fig. 2b) and by thermo-stability assays (Supplementary Fig. 8). Interestingly, eZiPro has a $T_m$ value of 50.9 °C which is two degrees higher than gZiPro and bZiPro, suggesting that binding of the 'TGKR' peptide to the catalytic site stabilizes the protease (Supplementary Fig. 8). To accurately probe the structural dynamics and distinguish

differences between these various protease constructs, we collected $^1\text{H-}^{15}\text{N-HSQC}$ NMR spectra for eZiPro, gZiPro and bZiPro free enzymes. These three constructs exhibit similar patterns, suggesting a shared overall fold in solution (Fig. 2b; Supplementary Fig. 9). The superimposed $^1\text{H-}^{15}\text{N-HSQC}$ spectra of eZiPro and bZiPro however, clearly show many distinct chemical shift changes between the two proteases (Fig. 2b). Several residues exhibited sharper peaks in the eZiPro construct. Based on the backbone assignment of eZiPro, these affected residues belong to both the NS2B cofactor and NS3 protease and are located at the active site (Figs 1 and 2a,b; Supplementary Figs 6 and 10). For example, residues H51 and S135 from the catalytic triad exhibit chemical shift perturbations and different peak intensities when we compared the spectra of eZiPro and bZiPro.These results confirm that in solution, the eZiPro active site is predominantly occupied by the 'TGKR' peptide. They also demonstrate that in the absence of substrate or inhibitors, the protease active site exchanges with the environment, causing the line broadening of these residues in bZiPro which has an empty active site.

**Table 1 | Data collection and refinement statistics.**

| | |
|---|---|
| *Data collection statistics* | |
| Wavelength (Å) | 1.0004 |
| Resolution range (Å) | 76.9-1.84 (1.9-1.84)* |
| Space group | P3₁ |
| Unit cell a, b, c, | 88.78 88.78 138.09 |
| α, β, γ (Å) (°) | 90.0 90.0 120.0 |
| Total number of reflections | 499,206 |
| Unique reflections | 105,911 |
| Multiplicity | 4.7 |
| Completeness (%) | 100 (100)* |
| $I/\sigma I$ | 14.1 (1.51)* |
| Wilson B-factor (Å²) | 22.2 |
| $R_{merge}$† | 0.058 (0.525)* |
| | |
| *Refinement statistics* | |
| Resolution range (Å) | 31.4-1.84 |
| $R_{work}$‡ (%) | 17.48 |
| $R_{free}$§ (%) | 20.20 |
| Number of non-hydrogen atoms | 6,819 |
| Macromolecules | 6,663 |
| Water | 936 |
| r.m.s.d. (bonds) (Å) | 0.005 |
| r.m.s.d.‖ (angles) (°) | 1.19 |
| Ramachandran favoured (%) | 96 |
| Ramachandran allowed (%) | 4 |
| Ramachandran outliers (%) | 0 |
| Clashscore | 12 |
| Average B-factor (Å²) | 28.80 |
| Macromolecules | 28.60 |
| Solvent | 42.90 |

* The numbers in parentheses refer to the highest resolution shell.
† $R_{merge} = \sum|I_j - \langle I \rangle|/\sum I_j$, where $I_j$ is the intensity of an individual reflection, and $\langle I \rangle$ is the average intensity of that reflection.
‡ $R_{work} = \sum||F_o| - |F_c||/\sum|F_c|$, where $F_o$ denotes the observed structure factor amplitude, and $F_c$ the structure factor amplitude calculated from the model.
§ $R_{free}$ is as for $R_{work}$ but calculated with 5% (3044) of randomly chosen reflections omitted from the refinement.
‖ r.m.s.d. = root mean square deviations.

Next, to find out whether the differences in structure dynamics between the three free enzymes also differentially affects inhibitor binding, we carried out binding studies between the protease constructs and BPTI—a potent serine protease inhibitor which has been extensively used to study NS2B-NS3 proteases from DENV and WNV (Fig. 2c–e). BPTI binds to bZiPro as indicated by chemical shift perturbations upon addition of the inhibitor (Fig. 2e). In contrast, eZiPro and gZiPro showed no obvious chemical shift changes upon BPTI addition, suggesting that these two proteases do not bind to the inhibitor significantly under the same experimental conditions (Fig. 2c,d). For eZiPro, 'TGKR' peptide occupies the active site in *cis* and strongly competes against BPTI. For gZiPro, the G₄SG₄ linker may cause steric hindrance and prevent BPTI from entering the active site.

**Enzymatic activities of different ZIKV protease constructs**. To gain insights into functional implications caused by the differences in the protease active site accessibility between the three constructs, we measured the protease activities of eZiPro, gZiPro and bZiPro using the peptide substrate Bz–Nle–Lys–Arg–Arg–AMC (Fig. 3a)[20]. The affinity values for substrate ($K_m$) are 2.086, 20.42 and 6.322 µM for gZiPro, eZiPro and bZiPro while the catalytic rates are 1.073, 1.79 and 5.302 AMC molecules per second per enzyme molecule respectively. Thus, bZiPro has the highest catalytic efficiency, approximately ten times higher than eZiPro, due to a three times higher affinity for substrate and three times faster turnover rate. Remarkably, the single-chain

enzyme gZiPro that is commonly used for screening inhibitors has both the lowest $K_m$ value and the slowest turnover rate. The results confirm that 'TGKR' peptide or the G₄SG₄ linker sterically hinders the protease from binding to BPTI. Comparison of ZIKV protease enzymatic activity against other flavivirus proteases is discussed in Supplementary Note 1.

The binding preference of BPTI for different constructs can be translated to differences in protease activity inhibition (Fig. 3b). The half maximal inhibitory concentrations (IC₅₀) of BPTI for eZiPro, gZiPro and bZiPro are 350, 76 and 12 nM respectively. In accordance with our NMR data, IC₅₀ values showed that BPTI inhibits eZiPro 30 times less efficient than bZiPro indicating that TGKR peptide competes with binding of BPTI to protease pocket by steric hindrance. BPTI inhibits gZiPro >6 times less efficiently compared with bZiPro. The higher IC₅₀ value for gZiPro is due to the artificial G₄SG₄ linker between NS2B and NS3, which hinders substrate from entering the active site.

## Discussion

In this study, we designed an eZiPro construct and solved its crystal structure which reveals the structure of protease in complex with the NS2B C-terminal peptide TGKR after cleavage. The protease adopts a closed conformation, which is similar to other flavivirus proteases (Fig. 1). This structure reveals that the two basic residues in the cleavage are critical for protease binding. Peptide inhibitors can be designed based on current structural information. NMR study also confirms that the protease adopts a closed conformation in solution while dynamic exchanges were found for the β-hairpin region (Fig. 2a and Supplementary Fig 6). N-terminal residues of NS3 are dynamic and fall off from the S' pocket, indicating their weak binding to the protease.

Our enzymatic assay reveals that the eZiPro and gZiPro exhibited lower enzymatic activity (Km and $k_{cat}$) than that of bZiPro. The C-terminal end 'TGKR' of the NS2B cofactor from ZIKV competes with substrate and hence decreases the catalytic activity of NS3 *in vitro*. In this respect, the present complex is reminiscent of the hepatitis C virus NS3-4A structure which was also captured following *cis*-cleavage at the junction between NS3 and NS4A (ref. 21). *Cis*-cleavage at the NS2B-NS3 junction as captured here for ZIKV protease was also reported to occur early during polyprotein processing for YFV or DENV (refs 22,23). Although it remains possible that the steric hindrance caused by the NS2B C-terminal tail has some regulatory role in virus replication in the infected cell, there is no evidence to suggest that there is an auto-inhibition mechanism to regulate the protease activity *in vivo*. This is different from the case of the gag and gag-pro-pol precursors of HIV where the last cleavage yields an auto-inhibited protease[24,25] or the alphavirus capsid protein which releases itself from the structural polyprotein precursor and remains inhibited with its own C-terminal end inserted in the protease active site[26]. In the future, pulse-chase experiments with anti-NS proteins sera, as conducted for YFV (ref. 27), are necessary to define the precise sequence and kinetics of proteolytic events in ZIKV infected cells.

Importantly, our results from enzymatic assay and NMR binding study of BPTI demonstrate that bZiPro is more suitable for inhibitor screening than gZiPro, because of its free active site accessible to substrate or inhibitors. gZiPro, on the contrary, appears less biologically relevant as the artificial linker is likely to introduce steric hindrance altering the substrate (inhibitor) binding kinetics. Our structural results reveal the importance of P1 and P2 residues in binding to protease and availability of S3 and S4 position in the pocket. These findings should inform the development of specific protease inhibitors against ZIKV, for

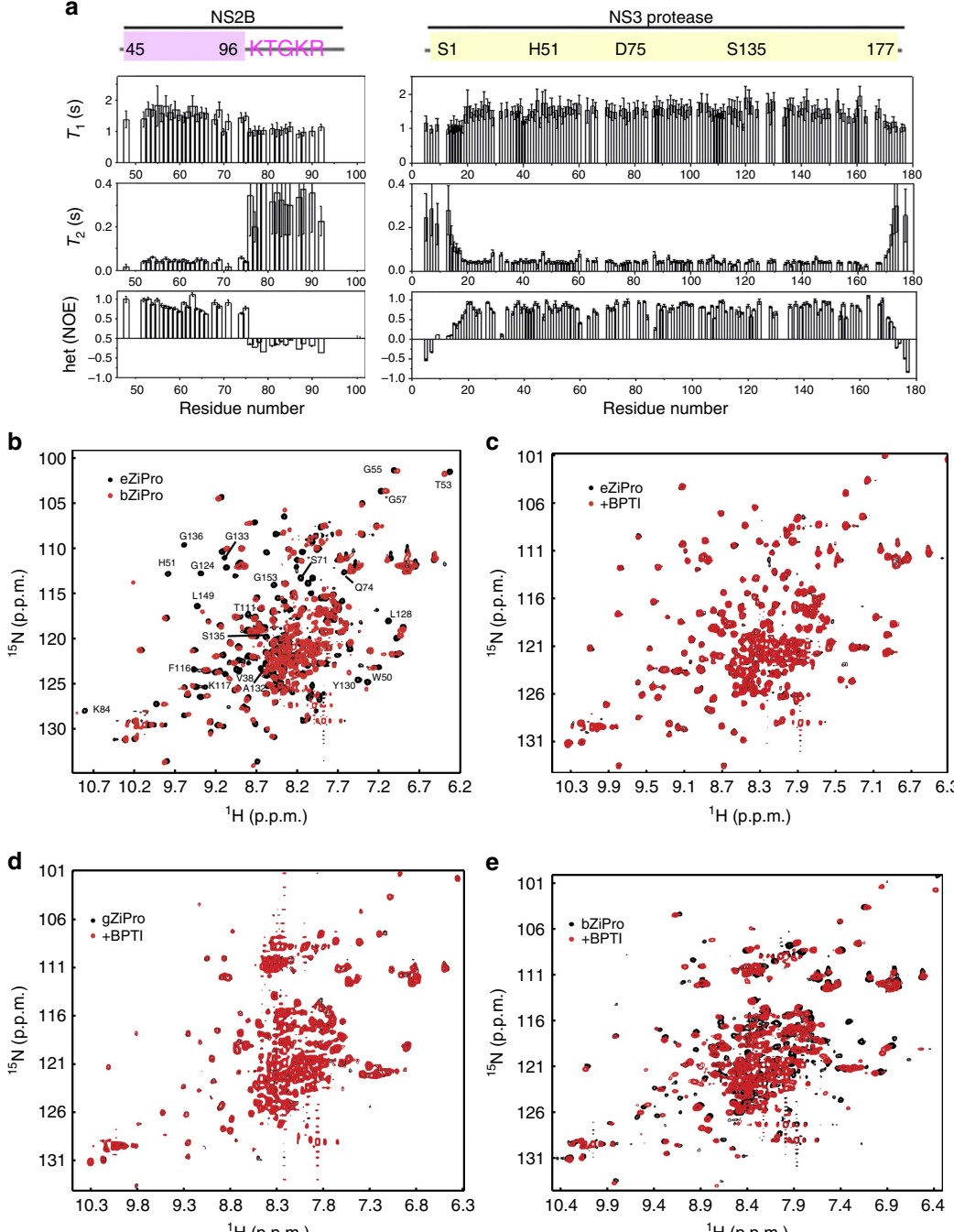

**Figure 2 | Molecular dynamics of the protease constructs in solution.** (**a**) Dynamics analysis of eZiPro. The $T_1$, $T_2$ and hetNOE values were plotted against residue number. Only assigned residues without signal overlapping were analysed. Error bars represent standard deviations from three values during curve fitting. The measurement was conducted on a magnet with proton frequency of 700 MHz at 310 K. (**b**) Overlaid $^1$H-$^{15}$N-HSQC spectra of eZiPro in black and bZiPro in red. (**c**) eZiPro, (**d**) gZiPro and (**e**) bZiPro before (in black) and after (in red) BPTI binding. Overlaid $^1$H-$^{15}$N-HSQC spectra of 0.5 mM protease in the absence and presence of 1 mM BPTI were collected. Only bZiPro exhibited clear interaction with BPTI.

instance by growing the P3 residue to reach more deeply into the S3 pocket.

While our manuscript was under review, Lei *et al.* published a crystal structure of ZIKV NS2B-NS3 protease (a gZiPro construct) in complex with a boronic compound cn-176 (ref. 28). Both the present manuscript and the Lei *et al.* study demonstrated that positively charged P1 and P2 residues of the substrate are critical for substrate binding whereas P3 and P4 appear less essential (Supplementary Fig. 3; Supplementary Table 1).

## Methods

**Expression constructs.** The codon optimized cDNA encoding NS2B-NS3 (GenBank Protein Accession number AEN75265.1) of ZIKV was synthesized (Genscript). The cDNA was cloned into the NdeI and XhoI sites of pET15b. The construct eZiPro contains the NS2B cofactor region (residues 45–96) linked with NS3 protein (residues 1–177) through the last five residues of NS2B (K126TGKR130). The gZiPro construct is similar except that the $G_4SG_4$ linker replaced K126TGKR130 of NS2B. The last construct is to express both NS2B cofactor region and NS3 protease in one vector-pETDUET-1 (Novagene) containing two promoter systems for expressing two proteins at the same time. NS3pro was cloned into the NcoI and HindIII sites of pETDUE-1 and the cDNA encoding NS2B cofactor region was cloned into the same vector using NdeI and XhoI sites.

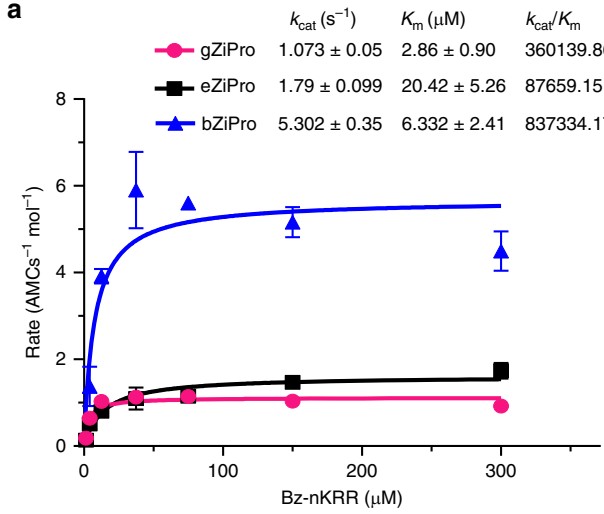

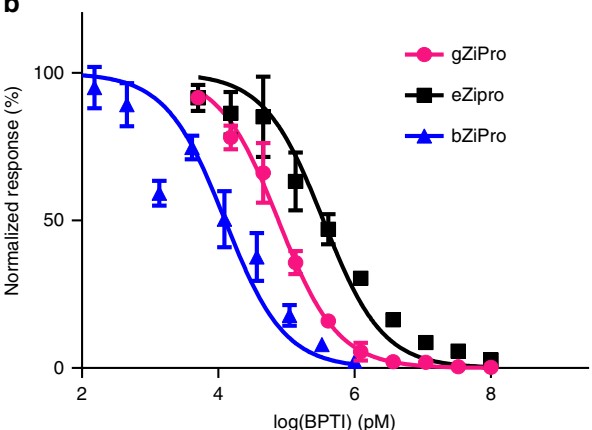

**Figure 3 | Enzymatic activities and BPTI bindings of protease constructs.**
(**a**) Protease activity of gZiPro, eZiPro and bZiPro were measured using the Bz—nKRR-AMC substrate. Assays were carried out as duplicates at 37 °C at 5 nM enzyme concentration with varying substrate concentrations ranging from 0 to 300 μM. Michaelis–Menten kinetics was plotted using non-linear regression function. Catalytic rates, binding affinity of substrate and catalytic efficiency are mentioned in the table inset the graph. Standard deviations for each data point are represented by error bars. The assays are carried out in duplicates or triplicates. (**b**) BPTI inhibition against Bz-nKRR-AMC substrate for gZiPro, eZiPro and bZiPro. The IC$_{50}$s for eZiPro, gZiPro and bZiPro are 350, 76 and 12 nM respectively. The assays are carried out in duplicates or triplicates. Standard deviations for each data point are represented by error bars.

**Protein expression and purification for NMR.** The plasmids harbouring different gene constructs were transformed into *E. coli* BL21(DE3) competent cells, respectively. Small scale induction was first carried out to examine recombinant protein expression. Cells were grown in 2 ml of Lysogeny broth (LB) medium. Protein expression was induced for 2–3 h at 37 °C by addition of β-D-1-thioga-lactopyranoside (IPTG) to 1 mM final concentration, when the optical density at 600 nm (OD$_{600}$) reached 0.8. Cells were harvested by centrifugation at 13,000$g$ for 1 min. SDS-PAGE loading dye was mixed with the cells and the mixture was heated at 100 °C for 15 min. The samples were resolved by SDS-PAGE and visualized by Coomassie blue staining. For large-scale protein production, the following procedures were used: Cells were grown in either the LB or M9 medium. Protein expression was induced overnight at 18 °C by addition of IPTG to 1 mM final concentration when the OD$_{600}$ reached 0.8. Cells were harvested by centrifugation at 11,000$g$ and 4 °C for 10 min. The cells were re-suspended in a buffer containing 20 mM sodium phosphate, pH 7.8, 500 mM NaCl, and 1.4 mM β-mercaptoethanol. Proteins were purified using immobilized metal affinity chromatography (IMAC) using Ni$^{2+}$-NTA resin and gel filtration chromatography using a HiPrep 26/60 Sephacryl S-200 column[29–31]. $^{13}$C, $^{15}$N, and $^2$H-labelled sample was prepared by growing bacterial cells in a M9 medium supplemented with $^{13}$C-glucose, $^{15}$NH$_4$Cl and D$_2$O. Protein expression was induced and purified as described above.

**Unlabelled protein expression and purification.** For crystallization, the following purification protocols were used: Cells were grown in LB broth enriched with 50 mM potassium phosphate buffer pH 7.4 and with 2.5% glycerol. Protein expression was induced overnight at 18 °C by addition of IPTG to 1 mM final concentration when the OD$_{600}$ reached 0.8. Cells were harvested by centrifugation at 3,470$g$ at 4 °C for 25 min. The cells were re-suspended in lysis buffer containing 20 mM phosphate buffer saline (PBS), pH 8.0, 500 mM NaCl, 2 mM β-mercap-toethanol, 10 mM imidazole and 5% glycerol. Cells were lysed by passing through a LM20 microfluidizer (Microfluidics, USA) at 20,000 psi and soluble fraction was obtained by centrifuging at 25,043$g$ for 1 h. Protein was purified using IMAC using Ni$^{2+}$-NTA beads (Qiagen, USA). After washing with lysis buffer containing 20–30 mM imidazole, protein was eluted with an elution buffer containing 300 mM imidazole. To remove the His-tag, Thrombin was added to the eluted fractions containing the protein and the mixture was dialyzed against the gel filtration buffer containing 20 mM Na-HEPES pH 7.5, 150 mM NaCl, 5% glycerol, 2 mM DTT for 20 h. After dialysis, the protein was further purified by size exclusion chromatography using a HiLoad 16/600 Superdex 75 pg column (GE Healthcare, USA). Fractions were collected and concentrated in a final buffer containing 20 mM HEPES pH 7.5, 150 mM NaCl, 5% glycerol, 2 mM DTT using spin concentrators (Vivaspin, Satorius, 10 kDa MWCO).

**NMR measurements.** Uniformly $^{15}$N-labelled protein was prepared in M9 medium supplied with 1 g l$^{-1}$ $^{15}$NH$_4$Cl. Protein was purified in a gel filtration buffer containing 20 mM HEPES, 150 mM NaCl, and 1 mM DTT. Although protein can be concentrated to more than 1 mM, a sample with 0.5–0.8 mM was used for data acquisition. The 2D $^1$H-$^{15}$N-HSQC spectra of purified proteases were acquired on a Bruker Avance II magnet equipped with a cryo-probe, with a proton frequency of 700 MHz. The data were acquired at both 25 and 37 °C. The pulse program used was from the Bruker pulse library (Topspin 2.1). The acquired data were processed with the NMRPipe software[32] and visualized with NMRView[33]. Proteins exhibited dispersed peaks at both temperatures. The spectrum was more dispersed at 37 °C, so only the results under this condition are shown.

Backbone resonance assignment was conducted using a $^{15}$N/$^{13}$C/$^2$H-labelled eZiPro protease in a buffer that contained 20 mM Na-HEPES, 150 mM NaCl and 1 mM DTT. Transverse relaxation-optimized spectroscopy (TROSY)[34,35]-based experiments that included $^1$H-$^{15}$N-HSQC, 3D-HNCACB, 3D-HNCOCACB, 3D-HNCOCA, 3D-HNCA, 3D-HNCACO, 3D-HNCO and NOESY-TROSY (with a mixing time of 100 ms) experiments. All the spectra were collected on Bruker Avance II 700 MHz equipped with a cryoprobe at 310 K. All the data were processed with NMRPipe[32] and Topspin (2.1), and visualized using NMRView[33] and CARA. Protein secondary structure were determined using TALOSN (ref. 36) and Cα chemical shifts[37].

The $^{15}$N- $T_1$ (spin-lattice relaxation time), $T_2$ (spin–spin relaxation time) and steady-state heteronuclear NOE (hetNOE) measurements[38] were acquired using the 0.8 mM triple labelled sample at 310 K. For the $T_1$ measurement, several 2D experiments with relaxation delays of 100, 200, 400, 600, 800, 1,200, 1,600, 2,000, 2,500 and 3,000 ms were recorded and processed. For the $T_2$ measurement, 2D spectra with relaxation delays of 17, 34, 51, 68, 85, 102, 119, 136 and 153 ms were collected and processed. The hetNOEs were obtained using two datasets that were collected with and without initial proton saturation for a period of 3 s (ref. 39).

**Protease assays.** The protease activity assays were carried out using benzol-Nle-Lys-Arg-Arg-aminomethylcoumarin (Bz-nKRR-AMC) (Peptide institute, Japan) modified from reference[20]. Bz-nKRR-AMC substrate with starting concentration of 300 μM was serially diluted two times in assay buffer (20 mM Tris–HCl, pH 8.5, 10% glycerol, 0.01% Triton X-100) and added to Corning 96 Well black plates with 5 nM protein diluted in the same buffer. For kinetics measurements, fluorescence readings were measured at 30 s interval for 5 min using Cytation 3 Mulitmode plate reader (BioTek) at an excitation wavelength ($\lambda_{ex}$) of 380 nm and an emission wavelength ($\lambda_{em}$) of 460 nm. Assays were carried out as duplicates or triplicates at 37 °C. To determine the amount of AMC released, a standard AMC curve was plotted with various concentrations of AMC. Initial velocities were calculated using the linear regression function in the GraphPad Prism software. Data were analysed and plotted using Michaelis-Menten equation with GraphPad Prism version 5.00 for Windows (GraphPad Software, San Diego California USA).

**Inhibition assays.** Inhibition of Zika proteases by BPTI was measured following the protocol reported by Leung *et al.*[40] with slight modifications. eZiPro, gZiPro and bZiPro were incubated at concentration of 5 nM in varying concentrations of BPTI from 100 to 0 μM in buffer containing 20 mM Tris pH 8.5, 10% glycerol, 0.01% Triton X-100 for 1 h at room temperature in Corning 96-well plate. The reaction was started by addition of Bz-nKRR-AMC substrate at 150 μM and initial velocities were read at 30 s intervals over 5 min using Cytation 3 Mulitmode plate reader (BioTek) at excitation wavelength ($\lambda_{ex}$) at 380 nm and emission wavelength ($\lambda_{em}$) at 460 nm. Assays were carried out as duplicates at 37 °C. BPTI concentrations were transformed into logarithmic values using Transform

function, Graph Pad Prism version 5.00 for Windows. Initial velocities were normalized to 0 to 100% using Normalize function in GraphPad Prism version 5.0. For normalization, initial velocities from wells with enzyme without substrate nor BPTI were used as 0% and those from wells with enzyme and substrate with 0 μM BPTI were used as 100%. IC$_{50}$ values were obtained from plotting normalized V$_i$ against log(concentrations) using Log(inhibitor) versus normalized response function, GraphPad Prism version 5.0 for Windows, GraphPad Software, La Jolla California USA, www.graphpad.com.

**Protein crystallization.** 1 μl of protein at a concentration of 40 mg ml$^{-1}$ was mixed with 1 μl of reservoir solution containing 0.2 M Na Malonate pH 4.0, 10% PEG 3350 by sitting drop vapour diffusion method. Crystals appeared at 18 °C after two days. Crystals were dehydrated overnight by soaking in 0.2 M Na Malonate pH 4.0, 20% PEG 3,350 and flash-frozen in liquid N$_2$ before mounting.

**Data collection and structure determination.** Diffraction intensities were recorded on PILATUS 2M-F detector at PXIII beamline at wavelength 1.0004 Å at the Swiss Light Source, Paul Scherrer Institut, Villigen, Switzerland. Diffraction intensities were integrated using iMOSFLM (ref. 41). Scaling and merging of the intensities were carried out using software POINTLESS and AIMLESS from CCP4 suite[29,42,43]. Data collection statistics are summarized in Table 1. A solution for the eZiPro crystal structure was determined by molecular replacement with the program MOLREP[44] using WNV NS2BNS3 protease structure as a search probe (PDB code 2FP7). The structure was subjected to iterative rounds of refinement using the Phenix.refine program[45–47] and manual rebuilding using Coot[48,49]. Significant twinning was detected giving an apparent hexagonal space group (P6$_{222}$ or the enantiomophic P6$_{422}$) instead of the true trigonal space group (P3$_1$) and the twin law ($-h, -k, l$) was applied during phenix refinement to correct for twining effects. Refinement statistics are summarized in Table 1. 96% of the residues are Ramachandran favoured regions, 4% in allowed region. Figures were generated using Pymol (ref. 50).

**Data availability.** The corresponding coordinates and structure factors are available from the Protein Data Bank (PDB) under accession code 5GJ4. NMR assignment data have been deposited in the Biological Magnetic Resonance Data Bank (BMRB) with accession number 26873. The authors declare that all other data supporting the findings of this study are available within the article and its Supplementary Information files, or are available from the authors upon request.

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

## Acknowledgements

We thank Drs Liew Chong Wai and El Sahili Abbas from Nanyang Technological University in Singapore, scientists from Swiss Light Source beam-line PXIII for their help with diffraction data collection. We thank Dr Subhash Vasudevan from Duke-NUS graduate medical school in Singapore for critically reading our manuscript. We thank Shuang Liu and Dr Alvin Hung from Experimental Therapeutics Centre, Agency for Science, Technology and Research (A*STAR) in Singapore for the thermal shift assay. This work was supported by (1) a start-up grant to D.L. lab from Lee Kong Chian School of Medicine, Nanyang Technological University, (2) National Medical Research Council grant CBRG14May051 to J.L., (3) A*STAR JCO grant (1431AFG102/1331A028) to C.K.

## Author contributions

Experiment design: C.K. and D.L. Cloning and protein work for NMR experiments: M.L., Y.R.L. and E.N. Protein work, crystallization, data collection, structure determination and analysis: W.P., Z.Z., Y.T. and D.L. NMR experiments: Y.L. and C.K. Enzymatic experiments: W.P. and Z.Z. Manuscript writing: J.L., C.K. and D.L. with inputs from all authors.

## Additional information

**Competing financial interests:** The authors declare no competing financial interests.

**Publisher's note**: 

