## [Peer Review File · Nature Communications]

REVIEWERS' COMMENTS:

Reviewer #1 (Remarks to the Author):

The authors have revised this manuscript taking into account my comments to the initial review, as well as those from the others. Now that I have read it in the light of the reviews from the three other referees, it seems clear to me that the inhibitory effect of the authentic C-terminal end of NS2B, which remains bound to the active site, is not demonstrated to happen *in vivo*; and since previous experiments with yellow fever virus (as pointed out by referee #3) show that the NS2B-NS3 cleavage takes place first, there is no point in asking whether there could be an ordered cleavage event as in the lentiviruses in which the NS2B-NS3 cleavage would be last, and then the protease would remain inhibited (I had asked that question initially, and now the authors have modified the paper to add it, whereas as pointed out by reviewer, in the case of flaviviruses it is more likely a regulatory event). In summary, the results are novel and I support publications, but the current version is a bit ambiguous, still leaving the possibility of an autoinhibition like in alphaviruses, which it appears not to be the case. Also, in that case, it is the C-terminal end of the capsid protein which remains in its active site, and therefore is clearly then autoinhibition, whereas here, it is the C-terminal end of another protein, NS2B, so it is not an auto-inhibitory event). Other than that, the manuscript is fine, and I strongly support publication.

Reviewer #2 (Remarks to the Author):

In this manuscript, which I reviewed earlier for a different journal, W Wint Phoo and coauthors report a study of the three-dimensional structure of Zika virus (ZKV) NS2B-NS3 protease. The crystal structure of the NS2B-NS3 protease complex was obtained at high resolution (1.84 Å). The structure reported in the paper represents the enzyme following the *cis*-cleavage at the NS2B-NS3 processing site. Importantly, in the crystal structure, the four C-terminal residues of the NS2B cofactor (TGKR-COOH) are found to remain bound at the protease active site, pointing to the possibility of auto-inhibitory function by the mature NS2B product.

The crystallographic and structural work appears indeed to be of high quality. Structures of the NS2B-NS3 protease from ZKV as well as from the closely related Dengue (DENV) and West Nile (WNV) flaviviruses have been reported before. Compared to the previous work, however, this manuscript does contain some important elements of novelty.

The most important point is that the structures of related DENV and WNV NS2B-NS3 complexes, as well as the recently reported ZKV NS2B-NS3 structure, were obtained by engineering a flexible linker connecting the core region of NS2B to NS3 in a single-chain protein, which remained uncleaved. In the present work, Wint Phoo and collaborators solved the crystal structure of a form of the ZIKV heterodimeric NS2B-NS3 protease in which the authentic NS2B-NS3 cleavage site is preserved and cleaved during the expression of the recombinant complex in bacteria. As mentioned above, the C-terminal portion of NS2B (Thr-Gly-Lys-Arg-COOH), generated from the cis-cleavage reaction, was observed to occupy the protease active site in the crystal structure, suggesting that this may represent a product-inhibited protease form. Importantly, the interaction observed in the present paper between the NS2B C-terminal tetrapeptide, containing the "di-basic" recognition motif, and the NS3 protease active site occurs primarily between the basic the P1 and P2 position of the substrate and the S1 and S2 substrate binding pockets, whereas S3 and S4 are largely unoccupied. This observation has valuable implications for design of potent and specific inhibitors of the ZKV protease.

The major issue that I raised upon reviewing the previous manuscript version was that – while the crystallographic structure points to the possibility of auto-inhibitory function by the mature NS2B product - there is no data in the paper indicating that this could indeed be a relevant biological phenomenon. Importantly, a similar observation was previously reported for the HCV NS3-NS4A protease (Yao et al. 1999, *Structure* 7, 1353). In this case, the C-terminal fragment of the NS3 protease, following the cis-cleavage of the NS3-NS4A precursor, was observed to remain bound at the protease active site in a putative "auto-inhibited" complex. However, for HCV NS3-NS4A, the C-terminal fragment generated by the cis-cleavage reaction does not appear to have an inhibitory or regulatory function in infected cells, indicating that not always interactions observed in crystal structures are biologically relevant. I do believe that the analogy with the HCV NS3-4A protease should be addressed more in detailed in the manuscript, including the possibility that what observed in the crystal structure may not be biologically relevant. The analogy now discussed by the authors - i.e., that of the alphavirus capsid protein - is interesting and appropriate. An even more pertinent example of auto-inhibited viral protease that the authors may want to comment is that of HCV NS2-NS3 protease (Lorenz et al. 2006, *Nature* 442, 831). Also in this case, the residues generated by the cleavage reaction remain coordinated in the enzyme active site predicting an inactive post-cleavage form. They authors should point out, however, to the fact that latter proteases (alphavirus capsid and HCV NS2-NS3) have evolved to cleaves only once and then become inhibited, which is not the case for ZKV NS2B-NS3 or HCV NS3-NS4A

In conclusion, I find that this is a very interesting paper containing some new information about an increasingly important potential target for antiviral drug discovery. Despite I find that the proposal of an auto-inhibitory role for the NS2B C-terminal peptide at the NS2B-NS3 junction following cis-cleavage is not fully supported by biological data, I do think that the timely publication of this work is important and that the overall quality of the manuscript justifies publication in *Nature Communications* after the suggestion above have been addressed.

Point-by-point response to the Referees' comments

Reviewer #1 (Remarks to the Author):

The authors have revised this manuscript taking into account my comments to the initial review, as well as those from the others. Now that I have read it in the light of the reviews from the three other referees, it seems clear to me that the inhibitory effect of the authentic C-terminal end of NS2B, which remains bound to the active site, is not demonstrated to happen *in vivo*; and since previous experiments with yellow fever virus (as pointed out by referee #3) show that the NS2B-NS3 cleavage takes place first, there is no point in asking whether there could be an ordered cleavage event as in the lentiviruses in which the NS2B-NS3 cleavage would be last, and then the protease would remain inhibited (I had asked that question initially, and now the authors have modified the paper to add it, whereas as pointed out by reviewer, in the case of flaviviruses it is more likely a regulatory event). In summary, the results are novel and I support publications, but the current version is a bit ambiguous, still leaving the possibility of an autoinhibition like in alphaviruses, which it appears not to be the case. Also, in that case, it is the C-terminal end of the capsid protein which remains in its active site, and therefore is clearly then autoinhibition, whereas here, it is the C-terminal end of another protein, NS2B, so it is not an auto-inhibitory event). Other than that, the manuscript is fine, and I strongly support publication.

Response: Thanks for the comments. We agree with the reviewer that there is no evidence to support the “auto-inhibitory” effect of the NS2B C-terminal tail to the protease *in vivo*. To avoid misguiding the readers, we therefore removed the related text from the discussion in our revised manuscript. See page 5 lines 40 to page 6 line 2.

Reviewer #2 (Remarks to the Author):

In this manuscript, which I reviewed earlier for a different journal, W Wint Phoo and coauthors report a study of the three-dimensional structure of Zika virus (ZKV) NS2B-NS3 protease. The crystal structure of the NS2B-NS3 protease complex was obtained at high resolution (1.84 Å). The structure reported in the paper represents the enzyme following the cis-cleavage at the NS2B-NS3 processing site. Importantly, in the crystal structure, the four C-terminal residues of the NS2B cofactor (TGKR-COOH) are found to remain bound at the protease active site, pointing to the possibility of auto-inhibitory function by the mature NS2B product.

The crystallographic and structural work appears indeed to be of high quality. Structures of the NS2B-NS3 protease from ZKV as well as from the closely related Dengue (DENV) and West Nile (WNV) flaviviruses have been reported before. Compared to the previous work, however, this manuscript does contain some important elements of novelty.

Response: Thanks for the comments.

The most important point is that the structures of related DENV and WNV NS2B-NS3 complexes, as well as the recently reported ZKV NS2B-NS3 structure, were obtained by engineering a flexible linker connecting the core region of NS2B to NS3 in a single-chain protein, which remained uncleaved. In the present work, Wint Phoo and collaborators solved the crystal structure of a form of the ZIKV heterodimeric NS2B-NS3 protease in which the authentic NS2B-NS3 cleavage site is preserved and cleaved during the expression of the recombinant complex in bacteria. As mentioned above, the C-

terminal portion of NS2B (Thr-Gly-Lys-Arg-COOH), generated from the cis-cleavage reaction, was observed to occupy the protease active site in the crystal structure, suggesting that this may represent a product-inhibited protease form. Importantly, the interaction observed in the present paper between the NS2B C-terminal tetrapeptide, containing the “di-basic” recognition motif, and the NS3 protease active site occurs primarily between the basic the P1 and P2 position of the substrate and the S1 and S2 substrate binding pockets, whereas S3 and S4 are largely unoccupied. This observation has valuable implications for design of potent and specific inhibitors of the ZKV protease.

Response: Thanks for the comments.

The major issue that I raised upon reviewing the previous manuscript version was that – while the crystallographic structure points to the possibility of auto-inhibitory function by the mature NS2B product - there is no data in the paper indicating that this could indeed be a relevant biological phenomenon. Importantly, a similar observation was previously reported for the HCV NS3-NS4A protease (Yao et al. 1999, Structure 7, 1353). In this case, the C-terminal fragment of the NS3 protease, following the cis-cleavage of the NS3-NS4A precursor, was observed to remain bound at the protease active site in a putative “auto-inhibited” complex. However, for HCV NS3-NS4A, the C-terminal fragment generated by the cis-cleavage reaction does not appear to have an inhibitory or regulatory function in infected cells, indicating that not always interactions observed in crystal structures are biologically relevant. I do believe that the analogy with the HCV NS3-4A protease should be addressed more in detailed in the manuscript, including the possibility that what observed in the crystal structure may not be biologically relevant. The analogy now discussed by the authors - i.e., that of the alphavirus capsid protein - is interesting and appropriate. An even more pertinent example of auto-inhibited viral protease that the authors may want to comment is that of HCV NS2-NS3 protease (Lorenz et al. 2006, Nature 442, 831). Also in this case, the residues generated by the cleavage reaction remain coordinated in the enzyme active site predicting an inactive post-cleavage form. They authors should point out, however, to the fact that latter proteases (alphavirus capsid and HCV NS2-NS3) have evolved to cleaves only once and then become inhibited, which is not the case for ZKV NS2B-NS3 or HCV NS3-NS4A.

Response: Thanks for the comments. We agree that there is no evidence to support the “auto-inhibitory” effect of the NS2B C-terminal tail to the protease *in vivo*. ZIKV NS2B-NS3 protease is similar to HCV NS3-NS4A protease but not HCV NS2-NS3 protease at both molecular level and at the viral infection level. We therefore removed the related text from the discussion in our revised manuscript (See page 5 lines 40 to page 6 line 2.) and suggest that the steric hindrance by the NS2B C-terminal tail may have some regulatory effect during virus replication and that it might be interesting to conduct some functional studies in the future.

In conclusion, I find that this is a very interesting paper containing some new information about an increasingly important potential target for antiviral drug discovery. Despite I find that the proposal of an auto-inhibitory role for the NS2B C-terminal peptide at the NS2B-NS3 junction following cis-cleavage is not fully supported by biological data, I do think that the timely publication of this work is important and that the overall quality of the manuscript justifies publication in Nature Communications after the suggestion above have been addressed.

Response: Thanks for the comments. We have followed the suggestion to change “auto-inhibition” to “steric hindrance” and revise the discussion accordingly.